# A Fucan Sulfate with Pentasaccharide Repeating Units from the Sea Cucumber *Holothuria*
*floridana* and Its Anticoagulant Activity

**DOI:** 10.3390/md20060377

**Published:** 2022-06-03

**Authors:** Zimo Ning, Pin Wang, Zhichuang Zuo, Xuelin Tao, Li Gao, Chen Xu, Zhiyue Wang, Bin Wu, Na Gao, Jinhua Zhao

**Affiliations:** School of Pharmaceutical Sciences, South-Central University for Nationalities, Wuhan 430074, China; zimo_ning1997@163.com (Z.N.); wangpin1994@163.com (P.W.); 18271682301@163.com (Z.Z.); txl6361@163.com (X.T.); gl15272975735@163.com (L.G.); 15171981226@163.com (C.X.); wangzhiyue0719@163.com (Z.W.); zhaojinhua@mail.scuec.edu.cn (J.Z.)

**Keywords:** sea cucumber, fucan sulfate, pentasaccharide, oligosaccharide, chemical structure, anticoagulant

## Abstract

A fucan sulfate (HfFS) was isolated from the sea cucumber *Holothuria*
*floridana* after proteolysis-alkaline treatment and purified with anion-exchange chromatography. The molecular weight (Mw) of HfFS was determined to be 443.4 kDa, and the sulfate content of HfFS was 30.4%. The structural analysis of the peroxidative depolymerized product (dHfFS-1) showed that the primary structure of HfFS was mainly composed of a distinct pentasaccharide repeating unit -[l-Fuc_2S4S_-α(1,3)-l-Fuc-α(1,3)-Fuc-α(1,3)-l-Fuc_2S_-α(1,3)-l-Fuc_2S_-α(1,3)-]*_n_*-. Then, the “bottom-up” strategy was employed to confirm the structure of HfFS, and a series of fucooligosaccharides (disaccharides, trisaccharides, and tetrasaccharides) were purified from the mild acid-hydrolyzed HfFS. The structures identified through 1D/2D NMR spectra showed that these fucooligosaccharides could be derivates from the pentasaccharide units, while the irregular sulfate substituent also exists in the units. Anticoagulant activity assays of native HfFS and its depolymerized products (dHf-1~dHf-6) in vitro suggested that HfFS exhibits potent APTT-prolonging activity and the potencies decreased with the reduction in molecular weights, and HfFS fragments (dHf-4~dHf-6) with Mw less than 11.5 kDa showed no significant anticoagulant effect. Overall, our study enriched the knowledge about the structural diversity of FSs in different sea cucumber species and their biological activities.

## 1. Introduction

Fucan sulfate (FS) is a kind of sulfated polysaccharide rich in l-fucose, which is mainly found in brown algae and marine invertebrates, including sea cucumbers and sea urchins. FSs have been widely studied and used in food, pharmaceutical, and cosmetic fields for their various biological functions, such as antidiabetic, antiobesity, antiviral [1], anti-inflammatory [2], anticoagulation [3], antitumor, antioxidation [4], and immune regulation activities [5]. These activities are usually highly dependent on physiochemical properties and the structures of FS, such as molecular weight (Mw), sulfate content, sulfated position, structural sequence, and chain conformation at a molecular level [3]. In particular, FSs from sea cucumbers are usually linear polysaccharides mainly consisting of α-L-fucose residues linked by (1,3), (1,4), and (1,2) glycosidic bonds, and the sulfated patterns are usually observed as 2-O-, 3-O-, 4-O-, and 2,4-di-O-sulfates. At present, many FSs from invertebrates consist of defined repeating units (Figure 1). For example, FSs from *Holothuria*
*fuscopunctata*, *Thelenota*
*ananas*, and *Stichopus horrens* are constructed through a single type of glycosidic linkage, mainly with one type of sulfated α-l-fucose residue [6]. Other FSs are composed of tetrasaccharide repeating units [7], such as FSs from *Acaudina*
*molpadioides* [8], *Holothuria*
*tubulosa* [9], *Isostichopus badionotus* [10], and *Pearsonothuria*
*graeffei* [11]. In contrast, FS isolated from *Stichopus*
*japonicas* is a branched polysaccharide and mainly consists of pentasaccharide repeating units [12]. FS from the sea cucumber *Holothuria*
*albiventer* is constructed with regular α (1,3) linked hexasaccharide repeating units [13]. The structural information is usually determined from the signal assignments in the one-dimensional/two-dimensional nuclear magnetic resonance (1D/2D NMR) spectra [14]. Most native FSs possess high Mw, and it is difficult to acquire high-quality NMR spectra. In addition, some minor signals, which could not be well assigned, may be from the structural heterogeneity of FS. Thus, the purified oligosaccharides are still required to provide solid and precise structural information. 

To obtain oligosaccharides, several methods used for glycosidic bond cleavage have been employed in natural FSs to yield products with a low degree of polymerization. Mild acid hydrolysis is considered a non-specific method for polysaccharide depolymerization, which has been used in FSs from sea urchins [15]. The results indicated that the glycosidic linkage between the non-sulfated fucose unit and the adjacent unit could be preferentially cleaved by acid, and the selective 2-desulfation was observed [16]. The available purified oligosaccharides with well-defined molecular sizes could achieve accurate sequence determination of native FSs.

FSs exhibit various pharmacological activities, and their anticoagulant activity has attracted great attention. Previous studies showed that FSs could prolong activated partial thromboplastin time (APTT) distinctly but could not affect prothrombin time (PT) and thrombin time (TT), which suggested that FSs exhibit anticoagulant activity mainly through inhibiting the intrinsic pathway of the coagulation cascade. The anticoagulant activity of FSs is highly dependent on the Mw and sulfate content [17]. Attempts to identify the relationship of the specific structural features of FSs with their anticoagulant activities were hindered due to their complex structures.

The sea cucumber *Holothuria*
*floridana* is a commercial species distributed in the Gulf of Mexico. The structure of FS has been reported, and it is mainly composed of a tetrasaccharide unit of [-3)-l-Fuc_2S4S_-α(1,3)-l-Fuc-α(1,3)-l-Fuc_2S_-α(1,3)-l-Fuc_2S_-α-(1,] by HILIC-ESI-HCD-MS/MS analysis. The heterogeneity was observed for the sulfate distribution in the FS chain [18]. However, the diverse structures observed from the MS/MS results likely do not support the presence of the tetrasaccharide unit. In this work, the FS was isolated from *H. floridana* (HfFS), and the physicochemical properties were studied. The primary structure was clarified using 1D/2D NMR analysis of the free radical depolymerized products, and a series of fucooligosaccharides were purified from the mild acid-hydrolyzed products of native HfFS. Moreover, the anticoagulant activities of native HfFS and several low-molecular-weight products were assessed through their effects on APTT, PT, and TT, which could reveal the relationship between the Mw and anticoagulant activity. Overall, our results provide the physicochemical properties and structural characteristics of HfFS and the effects of Mw on anticoagulant activities.

## 2. Results and Discussion

### 2.1. Purification and Physicochemical Properties of HfFS

The crude polysaccharide (Figure 1A) was extracted from the sea cucumber *H. floridana* by the method of papain digestion and alkaline hydrolysis with the yield of 7.24% by dry weight according to the previous method [19]. The crude polysaccharide was fractionated by precipitation with different concentrations of ethanol (EtOH), and the precipitate obtained with 60% of EtOH should be an FS-rich fraction as determined by high-performance gel permeation chromatography (HPGPC). Then it was further purified on an FPA98 strong anion-exchange column and eluted sequentially with gradient NaCl solutions. The native HfFS was collected in the water eluate, and its HPGPC profile showed a single chromatographic peak (Figure 1A), indicating its homogeneity. The Mw of HfFS was 443.4 kDa as calculated from the calibration curve as previously described [13]. The yield of HfFS extracted from dried sea cucumber body walls was about 0.96%. 

The monosaccharide composition of HfFS was qualitatively identified by reverse-phase high-performance liquid chromatography (HPLC), and the results were shown in Figure 1B, indicating that HfFS was composed mostly of L-fucose. The sulfate (SO_4_^2−^) content of HfFS was 30.4%, which was determined using ion chromatography [20] and calculated by the linear-fitting curve with a series of gradient concentrations of sulfate standard solution (Figure 1C, Appendix A). The molar ratio of sulfate ester to monosaccharide (l-Fucose) was about 0.67:1, suggesting that HfFS was composed of l-fucose with substitution of sulfate groups as observed in FSs from several other sea cucumbers [9,21]. The specific rotation of HfFS was −232.68°, which is in accordance with the l-configuration of fucose residues.

### 2.2. Infrared (IR) Spectroscopy Analysis

The structural characterization of HfFS was further analyzed by IR spectroscopy recorded in the region of 4000–400 cm^−1^, shown in Figure 1D. In general, the IR spectrum of HfFS was similar to those FSs from other sea cucumbers [22]. In brief, the bands at 3437 cm^−1^ and 3100–2900 cm^−1^ were from the characteristic O–H stretching vibrations and C–H stretching vibrations of carbohydrates, respectively. The H–O–H bending absorption of the associated water centered at 1641 cm^−^¹. Those bands observed at 1452 cm^−1^, 1385 cm^−1^, and 1346 cm^−1^ were attributed to the asymmetric and symmetric deformation vibrations of –CH_3_ of fucose residue. Three bands appeared at 1259 cm^−1^, 847 cm^−1^, and 581 cm^−1^ could be assigned to the S=O asymmetric stretching vibration, symmetric C–O–S stretching vibration, and S-O stretching vibration, indicating the presence of the sulfate groups. 

### 2.3. Structural Characterization of dHfFS-1 Prepared by Peroxidative Depolymerization

The 1D/2D NMR spectra of peroxidative depolymerized products (dHfFS-1) with Mw of ~8.7 kDa were recorded to analyze the primary structure of native HfFS. In the ^1^H NMR spectrum (Figure 2A red curve), five sets of signals were observed in the α-anomeric region, three of them were located at 5.30–5.47 ppm (residue A, D, E, respectively), and the other two were at 5.05–5.15 ppm (residue B and C, respectively), indicating that the residue A, D, and E should be sulfated fucoses, while residue B and C were the non-sulfated fucoses [16]. Additionally, the five types of residues showed basically equal integral areas in the ^1^H NMR spectrum (Figure 2A), suggesting that HfFS should be composed of pentasaccharide repeating units. Starting from the anomeric proton signals in the overlapped spectra of ^1^H-^1^H COSY, TOCSY, and ROESY (Figure 2B, Appendix A), the H2, H3, and H4 signals of each residue could be readily ascribed. The H5 resonances could be found from the methyl protons (H6) at 1.2–1.4 ppm, and the C6 was observed at 18.0–19.0 ppm. Moreover, the carbon signals (Figure 2C) could be assigned from the crosspeaks in the ^1^H-^13^C HSQC spectrum (Figure 2D, Table 1). In particular, the H-2 and C-2 chemical shift values of residues A, D, and E were shifted downfield obviously compared with those of non-sulfated fucose residues, indicating that these residues were 2-*O*-sulfated, respectively. Similarly, the O-4 position of residue A could generate the displacement toward the low field of H-4, indicating that this residue was sulfated at the *O*-4 position. Finally, residues A, D, and E could be deduced as 2,4-di-*O*-sulfated α-l-fucose (α-l-Fuc_2S4S_), 2-*O*-sulfated α-l-fucose (α-l-Fuc_2S_) and 2-*O*-sulfated α-l-fucose (α-l-Fuc_2S_), respectively, while B and C were non-sulfated α-l-fucose (α-l-Fuc). In addition, the glycosidic linkages in HfFS were analyzed by the ^1^H-^1^H ROESY spectrum (Figure 2B), and the correlation between H-1 and H-3 of these residues was identified. Thus, the presence of (1,3) linkages was confirmed in dHfFS-1. This evidence revealed that the sequence and linkage of the linear pentasaccharide unit was l-Fuc_2S4S_-α(1,3)-l-Fuc-α(1,3)-l-Fuc-α(1,3)-l-Fuc_2S_-α(1,3)-l-Fuc_2S_. Although 2D NMR can clarify the primary sulfation pattern and glycosidic linkage of fucose residues, further studies are needed to investigate the structural sequence of HfFS, especially the distribution of sulfate groups in chains. 

### 2.4. Mild Acid Hydrolysis of HfFS and Purification of Fucooligosaccharides

Mild acid hydrolysis of natural polysaccharides is a common and effective method to obtain oligosaccharides for structural analysis [16]. The hydrolysis process of HfFS in 10 mM trifluoroacetic acid (TFA) at 50 °C and 80 °C was studied, respectively. The Mw of the hydrolysates withdrawn at different times during 5 h was determined by HPGPC and was plotted with time (Figure 3). Obviously, the Mw of the sample decreased sharply in the first 0.5 h; then, the Mw was decreased slowly. The degradation process could be fitted by the non-linear formula, and the fitting formula at 50 °C was Mw = 48.8927 + 409.5013e^−1.2530*x*^ (*R*^2^ = 0.9599), in the case of 80 °C, the fitting formula was Mw = 2.1830 + 441.2296e^−8.5848*x*^ (*R*^2^ = 0.9999). The well-fitted profile indicated that the mild acid hydrolysis process of HfFS followed the first-order kinetics, which is in accordance with the degradation of linear polysaccharides [23]. Obviously, the rate of hydrolysis at 80 °C was higher than at 50 °C.

Then a large scale of native HfFS was subjected to mild acid hydrolysis in 10 mM TFA at 80 °C to produce depolymerized product (dHfFS-2) with a yield of 67%. The sulfate content was determined by ion chromatography (Appendix A) and calculated as 31.8%, which was similar to that of native HfFS, suggesting that the mild acid hydrolysis conditions used in this study did not cause obvious desulfation. Several independent peaks could be observed in the HPGPC profile (Figure 4A), indicating that dHfFS-2 contained a series of oligosaccharides with different molecular weights. For oligosaccharides purification, dHfFS-2 was loaded on the Bio-Gel P6 column, and five size-homogeneous fractions (F1, F2, F3, F4, and F5) were obtained. Particularly, F3 was further purified by strong anion-exchange chromatography (SAX) to yield F3-a and F3-b. The precise structures of these fractions were further elucidated by the 1D/2D NMR spectra assignments. Then, the structure of the native HfFS could be confirmed through this bottom-up strategy. 

### 2.5. Structural Determination of Fucooligosaccharides by NMR

FSs from marine invertebrates are generally composed of repetitive well-defined fucooligosaccharides building blocks, which could be decorated with different sulfation patterns in a species-specific manner [24]. Several FSs have been extracted and purified from various sea cucumbers, and their structures were usually identified by NMR spectra analysis. The FSs consisting of mono- or tetra-saccharide repeating units were mostly reported. Recently, three types of regular FSs [6,25,26,27] were found from *H**. fuscopunctata, T. ananas, S. horrens, S. herrmanni,* and *S. variegatus*, which have been identified as comprising mono-fucose repeating units with a specific pattern of sulfate substitution at C-2 or C-3 position and linked by α(1,4) or α(1,3) glycosidic bonds (Figure 1). The FSs composed of distinctive tetrasaccharide repeating units could be obtained from the body wall of sea cucumbers, including *L. grisea*, *I. badionotus*, *P. graeffei*, *H. tubulosa*, etc. [9,11,19,21,28]. Furthermore, branched FSs consisting of pentasaccharide repeating units were reported from *Stichopus*
*japonicus* [12], *Holothuria*
*edulis* [21], and *H. coluber* [29]. Overall, although FSs with the same structure may exist in different sea cucumber species, most of their specific sulfation patterns and the glycosidic linkages could vary from each other. 

**Scheme 1 marinedrugs-20-00377-sch001:**
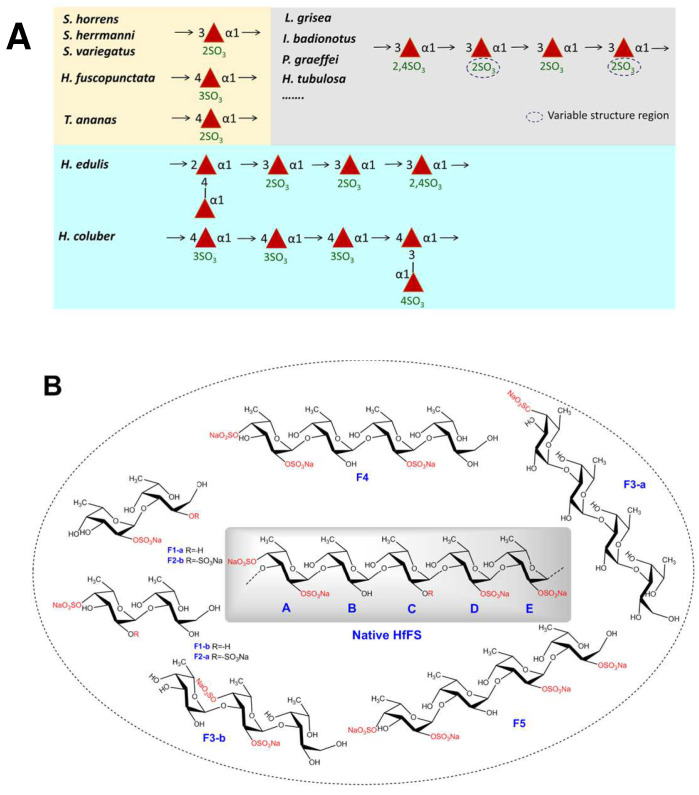
Typical repeating units of FSs found in sea cucumbers (**A**) and the structures of fucooligosaccharides in this study (**B**).

To our knowledge, the structure of FS in *H. floridana* has been revealed by HILIC-ESI-FT-MS or HILIC-ESI-HCD-MS/MS analysis of the sulfated oligosaccharides obtained by hydrothermal hydrolysis [18,30]. The structure of HfFS was first shown as -Fuc-Fuc_4S_-Fuc_2S4S_-Fuc_2S_-, then a tetrasaccharide unit of [-α-l-Fuc_2S4S_-(1,3)-α-l-Fuc-(1,3)-α-l-Fucp_2S_-(1,3)-α-l-Fuc_2S_-(1,3)-] were deduced, meanwhile, various oligosaccharide domains owing to the heterogeneous distribution of sulfate substitution were observed. However, the previous stucture description of HfFS was mainly on MS spectra or NMR spectra of native HfFS with high molecular weight. The purified oligosaccharides are still required to provide solid and precise structural information. Here, the primary structure of HfFS was revealed as a pentasaccharide repeating unit based on the full signal assignments of the depolymerized product of HfFS (dHfFS-1). Moreover, a series of fucooligosaccharides, including disaccharides, trisaccharides, and terasaccharides (Figure 1), were purified from mild acid-hydrolyzed HfFS (dHfFS-2), and their structural sequences were identified unambiguously by high-resolution NMR.

The purified fucooligosaccharides, including F1, F2, F3-a, F3-b, F4, and F5, were subjected to 1D/2D NMR spectra analysis. F1 was a mixture of disaccharides, and the main component F1-a was l-Fuc_2S_-α(1,3)-l-Fuc-ol, and the minor disaccharides were F1-b l-Fuc_4S_-α(1,3)-l-Fuc-ol and F1-c l-Fuc_2S4S_-α(1,3)-l-Fuc-ol. The α-anomeric protons (H1) of the non-reducing end fucose residues of these disaccharides were observed at 5.183, 5.006, and 5.209 ppm, respectively, suggesting the different sulfated types of these residues. Starting from the H1 of these residues in the ^1^H-^1^H COSY spectrum (Appendix A), the other protons could be assigned based on the spin-spin system. The reducing ends of these disaccharides were l-Fuc-ol. The ^13^C signals could be further assigned according to the ^1^H-^13^C HSQC spectrum (Appendix A). The detailed ^1^H and ^13^C signals assignment could be readily achieved and are shown in Appendix A. 

The ^1^H and ^13^C NMR spectra of F2 showed well-resolved signals, and together with the 2D NMR spectra, these signals could be assigned completely (Figure 5). F2 was composed of two disaccharides designated F2-a and F2-b with a ratio of 67% and 33%, respectively. In the ^1^H spectra, the signal located at 5.208 and 5.195 ppm could be ascribed to the α-anomeric protons of the non-reducing end fucose residues of F2-a (A) and F2-b (A’), respectively. From the signal, the other protons of A or A’ could be assigned based on the connectives in the ^1^H-^1^H COSY and TOCSY spectra (Appendix A, Table 2). Notably, the H-2 chemical shift values of residues A and A’ at 4.339 ppm shifted downfield by about 0.6 ppm compared with those of non-sulfated fucose residues, indicating that the sulfation substituted at O-2 of these residues. Meanwhile, the H-4 chemical shift of residue A indicated that it was sulfated at O-4. Thus, residue A in F2-a could be clarified as α-l-Fuc_2S4S_, and A’ in F2-b was α-l-Fuc_2S_. Regarding the reducing end of F2-a (B) and F2-b (B’), they were observed as the corresponding fucitol. The H-2 signal (4.558 ppm) of residue B’ indicated it was l-Fuc_2S_-ol, while B was l-Fuc-ol. The ^13^C signals could be fully assigned from the ^1^H-^13^C HSQC spectrum (Figure 5B), and the downfield shift of sulfated carbon confirmed the sulfated types of the fucose residues. Furthermore, the glycosidic linkages could be revealed according to the correlation peaks in ^1^H-^1^H ROESY and ^1^H-^13^C HMBC spectra (Appendix A, Figure 5B). The correlation between H-1 and H-3 of these residues, and the crosspeaks of H-1 of residue A and C-3 of residue B, H-1 of residue A’ and C-3 of residue B’ could be identified in the ^1^H-^1^H ROESY and ^1^H-^13^C HMBC, respectively, confirming that the presence of (1,3) linkages in F2-a and F2-b. Based on the above analysis, the compounds F2-a and F2-b could be deduced as l-Fuc_2S4S_-α(1,3)-l-Fuc-ol and l-Fuc_2S_-α(1,3)-l-Fuc_2S_-ol, respectively.

In the case of F3, the ^1^H NMR (Appendix A) showed several signals in the anomeric proton region, suggesting F3 may be composed of fucooligosaccharides with different structural sequences. Thus, F3 was subjected to SAX chromatography, and two fucooligosaccharides (F3-a and F3-b) were obtained (Appendix A). Interestingly, the 1D/2D NMR spectra analysis showed that F3-a was a tetrasaccharide (Appendix A, Figure 6A,B). The non-reducing residue (A) could be elucidated as α-L-Fuc_4S_, according to the high chemical shift values of H-4 (4.543 ppm) and C-4 (83.432 ppm) of this residue. Moreover, full assignments of the other residues (B, C, D) showed that they were all non-sulfated fucose residues. The glycosidic bonds could be confirmed by the crosspeaks observed in ^1^H-^1^H ROESY (Figure 6C) and ^1^H-^13^C HMBC spectra (Appendix A). These results affirmed that the structure of the major component of F3-a was α-l-Fuc_4S_-α(1,3)-l-Fuc-α(1,3)-l-Fuc-α(1,3)-l-Fuc-ol. The MS data (Appendix A) showed that F3-a is a monosulfated tetrasaccharide. In addition, a minor component was observed in the ^1^H NMR of F3-a (Figure 6A), and it could be identified as a monosulfated trisaccharide with a structure sequence of l-Fuc-α(1,3)-l-Fuc_2S_-α(1,3)-l-Fuc-ol (Appendix A). Regarding the structure of F3-b, three spin-spin coupling systems starting from the H-1 of each residue were clearly observed in the ^1^H-^1^H COSY spectrum (Appendix A), indicating that it was a compound with a trisaccharide structure. Additionally, distinct signals observed at around 1.1-1.2 ppm were the characteristic signals of methyl protons in the L-fucose residues with different sulfation patterns. Furthermore, according to the ^13^C and ^1^H-^13^C HSQC spectra (Figure 6E,F), all resonances were assigned readily and are shown in Appendix A. The resonance at 5.003 ppm and 5.260 ppm were assigned to α-anomeric protons of residue A at the non-reducing end and B in the middle of the chain, respectively, and the C-1 was at 100.66 and 100.47 ppm, respectively. In addition, the obvious downfield resonances of H-2 (4.465 ppm) and H-4 (4.750 ppm) of residue B indicated that it was sulfated at both C-2 and C-4, while residue A and the reducing end residue C were non-sulfated fucose. In the ^1^H-^1^H ROESY (Appendix A) and ^1^H-^13^C HMBC spectra (Figure 6F), the correlation between H-1 and H/C-3 of these residues was identified. Meanwhile, the position of glycosylation at O-3 was also observed by the downfield ^13^C resonance of the C-3 of residues B (75.08 ppm) and C (81.32 ppm), and the above analysis indicated the presence of (1,3) linkages in the fucan sulfate. The chemical formula (C_18_H_32_O_19_S_2_Na_2_) and molecular weight (662.53 Da) were determined based on the ESI-MS data (Appendix A). The structure of F3-b was demonstrated to be l-Fuc-α(1,3)-l-Fuc_2S4S_-α(1,3)-l-Fuc-ol.

Likewise, F4 and F5 could be deduced as tetrasaccharides linked by α(1,3) glycosidic bonds (Figure 7, Appendix A). Based on the full assignments of H/C chemical shifts of each residue (Table 3), the non-reducing residue was α-l-Fuc_2S4S,_ and the disaccharide in the middle chain was l-Fuc-α(1,3)-l-Fuc_2S_. In particular, the sulfation pattern of the reducing end fucitol was different in F4 and F5. In F4, this residue was the non-sulfated l-Fuc-ol, while it was l-Fuc_2S_-ol in F5. Thus, it could be inferred that the structure of F4 was l-Fuc_2S4S_-α(1,3)-l-Fuc-α(1,3)-l-Fuc_2S_-α(1,3)-l-Fuc-ol, and F5 was a tetrasaccharide with the structure of l-Fuc_2S4S_-α(1,3)-l-Fuc-α(1,3)-l-Fuc_2S_-α(1,3)-l-Fuc_2S_-ol.

Taken together, most of the oligosaccharides with well-defined structural sequences were in accordance with the major pentasaccharide repeating unit of native HfFS (Figure 1). The structures of tetrasaccharides F4 and F5 suggested that the residue C (L-Fuc) in the petasaccharide unit of native HfFS could be presented as l-Fuc_2S_, indicating the sulfation heterogeneity in HfFS. Moreover, the occurrence of 2-desulfation during the mild acid hydrolysis process of HfFS was observed, which has been reported in the studies of FSs from sea urchins and sea cucumbers [16]. 

### 2.6. Anticoagulant Activity Analysis

Fucan sulfate has exhibited various biological activities, and anticoagulant activity is widely concerned, which is highly associated with the structural features, especially sulfate content and molecular weight [3]. To evaluate the effects of Mw on anticoagulant activity, six low-molecular-weight products (dHf-1~dHf-6) were prepared from the native HfFS by mild acid hydrolysis. The Mw was determined by HPGPC and listed in Table 4, and their ^1^H NMR spectra are shown in Figure 8A. The effects of native HfFS and dHf-1~dHf-6 on the activated partial thromboplastin time (APTT), prothrombin time (PT), and thrombin time (TT) were assessed by plasma clotting assays [21]. The low-molecular-weight heparin (LMWH, Enoxaparin sodium) was used as a positive control. The results shown in Figure 8 and Table 4 indicated that the HfFS and its depolymerized products exhibit APTT-prolonging activities in a dose-dependent manner.

The concentrations of these compounds needed to double the APTT were calculated, and for the native HfFS (Mw 443.41 kDa), the concentration was 46.17 μg/mL, indicating that HfFS possesses intrinsic anticoagulant pathway inhibition activity. Interestingly, dHf-1 showed similar APTT-prolonging activity (42.49 μg/mL) to HfFS, suggesting that dHf-1 with Mw of 74.41 kDa could show full activity closed to the native HfFS. dHf-2 with Mw 27.68 kDa and dHf-3 with Mw 20.32 kDa showed APTT-prolonging activity at 82.63 and 95.43 μg/mL, respectively, which was weaker than dHf-1, indicating the anticoagulant potencies of HfFS showed a certain correlation with their Mw. However, the effects of HfFS and dHf-1~-dHf-3 on APTT were much weaker than LMWH (9.60 μg/mL). Notably, dHf-4~dHf-6 with Mw lower than about 11.5 kDa showed negligible inhibitory activity on the intrinsic anticoagulant pathway. Moreover, the HfFS and its depolymerized products had no significant or weak effects on PT or TT-prolonging activities at the concentrations tested. 

## 3. Materials and Methods

### 3.1. Materials

The dry body walls of the sea cucumber *Holothuria*
*floridana* were purchased from Guangzhou, Guangdong province, China. Amberlite FPA98Cl ion exchange was purchased from Rohm and Haas Company (New Iberia, LA, USA). Bio-Gel P6 and P2 were from Bio-Rad Laboratories. Sephadex G-10, G-25, and G-100 were from GE Healthcare Life Sciences (USA). Deuterium oxide (D_2_O, 99.9% Atom D) was obtained from Sigma-Aldrich. The activated partial thromboplastin time (APTT) kits, prothrombin time (PT) assay kits, thrombin time (TT) assay kits, CaCl_2,_ and standard human plasma were obtained from MDC Hemostasis (Berlin, Germany). All other chemicals were of reagent grade and obtained commercially.

### 3.2. Extraction and Purification of HfFS

The crude polysaccharide was extracted from the body walls of *H. floridana* according to the method in our previous report [19]. The dried body walls (1805.1 g) were crushed and incubated in 18 L H_2_O with 0.1% (*w*/*v*) papain at 55 °C for 6 h. Then it was treated with 0.5 M NaOH at 60 °C for 2 h. The product was cooled to room temperature, neutralized, and centrifuged (4000 rpm, 15 min). The protein in the supernatant was precipitated by adjusting the pH to 2.8 with 6 M HCl. The sediment was removed by centrifugation, and ethanol was added to the supernatant with the final concentration of 60% (*v*/*v*) to yield the crude polysaccharide. The crude polysaccharide was dissolved in deionized water and decolorized with 3% (*w*/*v*) H_2_O_2_ at 50 °C for 2 h (pH = 10). It was precipitated by ethanol with the final concentration of 60% (*v*/*v*), and half of the sediment was purified by FPA98 strong anion-exchange chromatography, eluted sequentially with H_2_O, 0.5 M, 1.0 M, 1.5 M, and 2 M NaCl. The H_2_O eluted fractions were collected and dialyzed against water by ultrafiltration with a 3500 Da molecular weight cut-off membrane. The retentate was lyophilized, and the purified polysaccharide HfFS was obtained. The homogeneity of HfFS was detected by the HPGPC method. 

### 3.3. Preparation of the Depolymerized Products dHfFS-1 by Peroxidative Depolymerization

The low-molecular-weight HfFS was prepared by peroxidative depolymerization induced by H_2_O_2_ in the presence of copper (II) [31,32]. A total of 50 mg of native HfFS and 1 mg of copper (II) acetate monohydrate were dissolved in 2 mL of deionized water, then 200 μL of 10% (*w*/*v*) H_2_O_2_ was added and incubated at 35 °C for 4 h. The reaction was terminated by adding 3 mg of EDTANa_2_. Then, 4 mL saturated NaCl solution was added, and the mixture was precipitated with ethanol (90%, *v*/*v*) and kept at 4 °C overnight. After centrifugation (4700 rpm, 15 min), the precipitate was collected and then dissolved in deionized water and repeated the above steps three times. The derivatives were desalted by Sephadex G-10 and lyophilized to yield dHfFS-1. 

### 3.4. Mild Acid Hydrolysis of HfFS

The HfFS was depolymerized with diluted trifluoroacetic acid by the previous method [33]. HfFS (3.08 g) was dissolved in 300 mL of 10 mM trifluoroacetic acid and hydrolyzed at 80 °C for 5 h. The reaction was neutralized with 6 M NaOH, and NaBH_4_ was added to reduce the hemiacetal groups at the reducing ends of oligosaccharides at 50 °C for 2 h. Finally, the depolymerized product (dHfFS-2) was obtained after neutralizing, desalting (Sephadex G-25 column), and lyophilizing.

For further evaluation of the effect of the Mw on anticoagulant activities, 306.6 mg of HfFS was subjected to mild acid hydrolysis at 50 °C according to the above steps. The hydrolysates were taken out at 5, 6, 7, and 8 h and neutralized, respectively. Then these samples were fractionated on a Sephadex G-100 column. Based on the HPGPC results of the eluted fraction, six size-homogeneous fractions (dHf-1~dHf-6) were obtained, and the Mw was measured as the below section of “Determination of weight-average molecular weight (Mw)”.

### 3.5. Purification of Fucooligosaccharides from dHfFS-2

Based on the different Mw of the oligosaccharides in dHfFS-2, dHfFS-2 was fractionated by repeated gel permeation chromatography (GPC) using Bio-Gel P6 column (fine, 2 × 180 cm, Bio-Rad). Briefly, dHfFS (~0.65 g) was dissolved in 3 mL of deionized water, subjected to the Bio-Gel P6 column equilibrated with 0.2 M NaCl, and then eluted with the same solution at a flow rate of 10 mL/h, and about 100 fractions (2 mL/tube) were collected. Each fraction was measured by the sulfuric acid-phenol method at 482 nm. According to the eluted profile, some fractions were analyzed by a Superdex Peptide 10/300 GL column (GE Healthcare Life Sciences), eluted with 0.2 M NaCl solution at the flow rate of 0.4 mL/min, monitored by a RID. The fractions that showed a single peak were combined and desalted by Sephadex G-10 or Bio-Gel P2 column and lyophilized, respectively. The obtained components were designated as F1, F2, F3, F4, and F5, respectively. The complex signals presented in the ^1^H NMR of F3 were further purified using a Spherisorb SAX column (4.6 × 250 mm, Waters). The column was equilibrated and eluted with 0.05 M KH_2_PO_4_, pH 3.0, at a flow rate of 1.0 mL/min. The elution profile was measured by RID, and three peaks were collected, desalted, and freeze-dried. Two homogeneous oligosaccharides (F3-a and F3-b) were obtained for further structural analysis. 

### 3.6. Kinetics of Mild Acid Hydrolysis of HfFS at Different Temperatures

In the previous studies of the chemical structure and structure-activity relationship of FSs from natural sources, mild acid hydrolysis has become a well-known depolymerization method to obtain oligosaccharides [16]. The kinetics of the mild acid hydrolysis process for HfFS were studied. In this part, HfFS (300 mg each) was dissolved in 10 mM trifluoroacetic acid (30 mL) and depolymerized at 50 °C and 80 °C, respectively. Samples (3 mL) were taken out every half an hour for 5 h and neutralized to pH 7 immediately with 6 M NaOH. All the samples were desalted by Sephadex G-10 and lyophilized. Molecular weights were analyzed by the “Determination of weight-average molecular weight (Mw)” method. 

### 3.7. Chemical Composition and Physicochemical Properties of HfFS

**Monosaccharide composition analysis.** The monosaccharide composition of HfFS was determined by a sensitive high-performance chromatography (HPLC)-based method [34,35]. Briefly, HfFS (2.0 mg) was hydrolyzed completely in 2 mL trifluoroacetic acid (2 M) at 110 °C for 4 h, and then the hydrolysates were labeled with 1-phenyl-3-methyl-5-pyrazolone (PMP) in alkaline solution. The labeled monosaccharides were separated by HPLC using an Agilent Eclipse XDB C18 column (150 mm × 4.6 mm), monitored by UV absorbance at 250 nm. 

**Content of sulfate groups.** The sulfate content of the polysaccharide was analyzed by ion chromatography (IC). HfFS (6.2 mg) or dHfFS-2 (4.2 mg) was hydrolyzed in 2 M trifluoroacetic acid at 110 °C for 8 h. The final dried hydrolysates were dissolved in 50 mL of H_2_O. Certified 100 mg/L potassium sulfate was diluted to yield sulfate standards solutions with sulfate concentrations of 10, 30, 50, 70, and 90 mg/L, respectively. Then, use the Dionex ICS-2100 ion chromatography system (Thermo Fisher Scientific, USA) to determine the sulfate content of the standards and samples, draw the sulfate content curve of the standards according to the results, and calculate the sulfate content of HfFS or dHfFS-2. For IC, the concentration of sulfate was determined from the area versus added analyte mass, both corrected by sample weight. 

**Determination of weight-average molecular weight (Mw).** The Mw of HfFS, the peroxidative depolymerized product (dHfFS-1), the mild acid-hydrolyzed products for kinetics studies, and the products for anticoagulant activity assays were calculated as previously described [6]. The sample was dissolved in deionized water and prepared as 10 mg/mL. An Agilent Technologies 1260 series apparatus (Agilent, USA) equipped with RID and a Shodex OH-pak SB-804 HQ column was used. This column was calibrated by the dextran standards D_0_-D_8_ and D_2000_ with Mw of 180, 2500, 4600, 7100, 10,000, 21,400, 41,100, 84,400, 133,800, and 2,000,000 Da, respectively. The logMw vs. Ve of dextran standards were plotted as the original standard curve, and it was corrected by data of dHSG-2 (27,760 Da) or dHSG-5 (5297 Da). dHSG-2 and dHSG-5 were the low-molecular-weight products of the fucosylated glycosaminoglycan from *H. fuscopunctata* prepared through the β-eliminative depolymerization method, and their Mw and number-average molecular weight (Mn) were determined by multi-angle laser scattering-SEC method [36]. 

**Optical rotation.** The specific rotation of HfFS (2.0 mg/mL in H_2_O) was determined by using Jasco p-1020 digital polarimeter with a sodium monochromatic source at the detection temperature of 20 °C. 

### 3.8. IR and NMR Spectra Analysis

**IR analysis.** According to the previous method [22], HfFS (2 mg) was dried under vacuum at 40 °C for 24 h and then pressed pellet with KBr. The FT-IR spectrum was recorded on a Nicolet iS10 FT-IR spectrometer (Thermo Fisher Scientific, Waltham, MA, USA) in the wavelength range of 4000–400 cm^−1^.

**NMR analysis.** The ^1^H NMR analysis of samples dissolved in 0.5 mL D_2_O was performed on a Bruker Avance 600 MHz spectrometer (Bruker, Rheinstetten, Germany) equipped with a ^13^C/^1^H dual probe in FT mode at 298 K. The ^13^C, 2D NMR analysis items including ^1^H-^1^H correlated spectroscopy (COSY), total correlation spectroscopy (TOCSY), rotating frame overhauser effect spectroscopy (ROESY), ^1^H-^13^C heteronuclear single-quantum coherence (HSQC) and heteronuclear multiple bond coherence (HMBC) of oligosaccharides (F1 and F2) were recorded on a Bruker Avance III 600 MHz spectrometer. In addition, the spectra of F3-a, F3-b, F4, F5, and dHfFS-1 were recorded on a Bruker Avance 800 MHz spectrometer. The detailed experimental conditions of NMR acquisitions have been shown in the supporting information.

### 3.9. Mass Spectrometry Analysis

Negative-ion ESI-MS spectrometry analysis of F1 and F2 was carried out on an Accurate-Mass Q-EXACTIVE LC/MS spectrometer (Thermo Technologies, Walthamcity, MA, USA). Each oligosaccharide was dissolved in purified water, typically at a concentration of 100 µg/mL. Mass spectra were acquired in the negative-ion mode under the following conditions: spray voltage 3800 V, capillary temperature 325 °C, auxiliary device temperature 380 °C. The MS spectra of the oligosaccharides were acquired in scan mode (*m*/*z* scan range 400–2000). In particular, the MS spectra of F3-a, F3-b, F4, and F5 were acquired on a 6540 UHD Accurate-Mass Q-TOF LC/MS spectrometer (Agilent Technologies, Santa Clara, CA, USA) according to previous methods [33]. 

### 3.10. Anticoagulant Activities of HfFS and Its Low-Molecular-Weight Products In Vitro

The anticoagulant activities of HfFS and six depolymerized samples (dHf-1~dHf-6) with different molecular weights (74.7 kDa, 27.7 kDa, 20.3 kDa,11.5 kDa, 8.3 kDa, 6.0 kDa) were evaluated by plasma clotting assays as described [36], including activated partial thromboplastin time (APTT), prothrombin time (PT) and thrombin time (TT). The commercial APTT, PT, and TT reagents and standard human plasma were used to measure APTT, PT, and TT on a coagulometer (TECO MC-4000, Berlin, Germany). Low-molecular-weight heparin (LMWH) was used as a positive control. The concentrations of samples in plasma were plotted with clotting times in a linear-fitting method, and the concentration required for doubling clotting time was calculated from the fitting curves. Determinations were performed in duplicate.

## 4. Conclusions

In this study, a fucan sulfate (HfFS) was extracted from the sea cucumber *Holothuria*
*floridana* by papain digestion-alkaline hydrolysis and purified by anion-exchange chromatography. The Mw of HfFS was determined to be 443.4 kDa by HPGPC. Chemical composition analysis showed that HfFS was composed of fucose and sulfate, and the sulfate content was 30.4%. The structural analysis of HfFS was mainly based on 1D/2D NMR spectroscopy of the peroxidative depolymerized product (dHfFS-1). Particularly, a “bottom-up” strategy was employed to investigate the structure of HfFS. Firstly, the effects of reaction temperature on the molecular weight during the mild acid hydrolysis of HfFS were investigated, and the kinetic model for its degradation indicated that mild acid hydrolysis followed the first-order reaction. A series of fucooligosaccharides (disaccharides, trisaccharides, and tetrasaccharides) were purified from the mild acid-hydrolyzed product of HfFS, and their structures were identified through 1D/2D NMR spectra. These results confirmed that HfFS was mainly composed of a distinct pentasaccharide repeating unit -[l-Fuc_2S4S_-α(1,3)-l-Fuc-α(1,3)-Fuc-α(1,3)-l-Fuc_2S_-α(1,3)-l-Fuc_2S_-]*_n_*-. Furthermore, the sulfation heterogeneity in HfFS was also observed from the structural sequence of fucooligosaccharides. In addition, anticoagulant activity assays of native HfFS and its depolymerized products (dHf-1~dHf-6) in vitro suggested that HfFS exhibited potent APTT-prolonging activity, and the potencies were decreased with the reduction in molecular weights. Interestingly, dHf-4~dHf-6 with Mw less than 11.5 kDa showed no anticoagulant effect. Overall, our study enriched the knowledge about the structural diversity in different sea cucumber species and their biological activities.

## Data Availability

All data generated or analyzed are available from the corresponding author on request.

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
