# Peer review of "A Fucan Sulfate with Pentasaccharide Repeating Units from the Sea Cucumber Holothuriafloridana and Its Anticoagulant Activity"

_marinedrugs, 2022, doi:10.3390/md20060377_

Round 1
Reviewer 1 Report
This paper analyses the structure of a fucan sulfate from the sea cucumber Holothuria floridana. The NMR analysis of the peroxidative depolymerised polysaccharides and oligosaccharides generated by mild acid hydrolysis is comprehensive.
The major issue with this paper concerns the characterisation of the depolymerised HfFS and the oligosaccharides generated by mild acid hydrolysis. The native HfFS has sulfate:fucose ratio of about a 1.1:1, but there is no data to show if this is changed by either peroxidative depolymerisation or mild acid hydrolysis. Thus, it is difficult to say that “The 1D/2D NMR spectra of peroxidative depolymerized product (dHfFS-1) were recorded to analyze the primary structure of native HfFS.” (line 129). Please address this.
Also, you show the molecular weight of dHfFS-1 in Fig. 1A, but do not seem to mention it in the text – please include this.
The paper requires editing for language usage. I suggest that you use an English language expert to polish your paper, particularly in the use of the grammar and tense (i.e. present/past tense in the same sentence).
The carbohydrate nomenclature needs attention: ensure that in α-L-fucose, L is a smaller font size. Also, α (1→3), should be α-(1→3).
Further comments:
Line 39 should be “chain conformation”
Line 42: change to: “…observed as 2-O-, 3-O-, 4-O- and 2,4-di-O-sulfates.
Line 42: change to: Many FS from invertebrates consist of defined repeating units.
Line 45: I do not understand the end of this sentence. Perhaps it is better to start a new sentence, “Other FS are (be careful not to mix tenses) composed of….”
Line 47: you already say “such as” earlier in this sentence so do not use “etc.”, simply finish the sentence “…P. graeffei [11].
Also, start the next sentence with, “In contrast,..” rather than “However,..”
Line 73: use commercial, rather than economical.
Line 74: delete “In fact,..”
Line 103: suggest to say “..that HfFS was composed mostly of L-fucose.”
Line 132: Is there a rational for assigning the anomeric resonances A-E in the order you have; it seems strange not to label them A to E from low to high field? Also, on what basis have you assigned the sulfated and non-sulfated fucose residues and is there literature to cite here?
For analysis of the oligosaccharides, labelling of the X-axis in Fig. S1 is ambiguous and it seems that the chemical shifts for A1 and A’1 shown in Table S1 are incorrect – please improve this. Also, state the differences in the assignments for the 2-sulfated and 4-sulfated species.
For fraction F2 there is a mistake in Table 2, where H-1 of the disulfated oligosaccharide should be 5.208 according to Fig. 5. Fraction F2-a appears to be identical to fraction F1-c, but has very different chemical shift assignments. Can you explain this?
The rationale for fractionating F3 by SAX chromatography is not clear and the methodology for this separation is only mentioned briefly. Please give sufficient details of the method and show the chromatogram in the supplementary material. At line 229, refer to Table S2 (ESI-MS data) as further evidence that F3-a is a tetrasaccharide; also, why do you not refer to Fig. 6A here? It seems from Table S3 that F3-a has two components (a major monosulfated tetrasaccharide and a minor major monosulfated trisaccharide), but there is no mention made of the latter in the text. With regards to the chemical shifts for the anomeric resonances of non-reducing 4-sulfated Fuc of F3-a-I (4.999/98.48 ppm), would you expect such differences when compared with those of the non-reducing 4-sulfated Fuc of F1-b (5.124/100.37 ppm)?
Line 235: be specific and say that this is the structure of the major component of F3-a.
Line 239: include reference the MS data (Table S2) showing that F3-a is a disulfated trisaccharide.
I am not sure if the structure of the paper is correct; I would expect to see the Conclusions before the Materials and Methods.
Line 364: ensure that you include 60 % v/v for the ethanol concentration and that the hydrogen peroxide concentration is given correctly.
Line 369: ensure that you say that you dialysed against deionised water.
Line 379: “…with ethanol (90%, v/v), and kept at 4 °C overnight.”
Line 386: “The reaction was neutralized with NaOH (how much of what concentration), NaBH4 was added to reduce the…”
Line 389: how was the sample desalted?
Line 397: please provide more details of the fractionation of the oligosaccharides by gel filtration and SAX chromatography.
Author Response
Dear reviewer,
We would like to show our great appreciation for your comments and suggestions on our manuscript. The manuscript has been modified accordingly, and detailed corrections are listed point by point in the attached file.

Reviewer 2 Report
With a relative anticoagulant activity evaluation, the present study exploits the structural characterisation of polysaccharide repeating units of sulphated fucan extracted from Holothuria floridana.
The authors should revise the English and measurement units in the main paper for the presence of many typos, repetitions, and redundancy. The manuscript is acceptable for publishing after major revisions, as it lacks unity, and the results are confusedly described. It seems to be written by many hands.
Moreover, several issues should be considered in graphs and modified carefully by the authors.
There are specific problems as follows:
Introduction:
Page 2, lines 45-49 = The authors should type the name of the invertebrates with their full names since they are introduced for the first time. Moreover, they should describe the chemical structure of the repeating unit, as done for H. Floridana.
Page 2, lines 57-64 = The authors should report state of the art for the several methods used for the depolymerisation of fucans and their effect on the sulfation pattern of polysaccharides.
Page 2, line 75 = Please use round brackets for glycosidic bonds in the structural formula.
Page 2, line 81 = The authors report the use of free radical depolymerisation, which should also be stated in the abstract. Please uniform.
Results and discussion:
Page 2, lines 94-95 = The authors should move “Shodex OH-pak SB-804 HQ column (8 mm x 300 mm)” in the “Materials and methods” section.
Page 3, Figure 1 = The authors, after, HPGPC chromatograms, describe in the main text HPLC ones; therefore, they should give consistency and uniformity, naming HPLC chromatograms as B, then IR as C and so on.
Page 3, Figure 1 A = The authors should use different colours for different curves.
Page 3, Figure 1 D = The authors should zoom the calibration curve and preferably move to the "Supporting Info".
Page 3, lines 119-126 = The authors should reorder the discussion about bands assignment, starting from 4000 to 400 cm-1. Moreover, please assign all the bands detected.
Page 4, line 131 = Please state “red curve” in Figure 2 A.
Page 5, Figure 2 B = The authors display this figure without describing it in the main text. They only report the C-5 chemical shift of dHfFS-1. Please remove the figure or discuss it in the text.
Page 5, Figure 2 C = Please change the figure with the un-zoomed one for more evident evidence of correlations.
Page 5, line 170 = The authors should move “Shodex OH-pak SB-804 HQ column (8 mm x 300 mm)” in the “Materials and methods” section.
Page 5, Paragraph 2.4 = The authors state the mild acid hydrolysis at 50°C, but the obtained result is not discussed, differently from the one conducted at 80°C.
Page 6, line 194 = The authors report the F1 fraction, which is a mixture of oligosaccharides; therefore, they should state and discuss F1-a and F1-b, as written in Scheme 1.
Page 6, lines 196-197 = The authors should discuss NMR spectra reported in Figure S1, as done for other fractions.
Page 7, line 208 = Please change “Table 1” with “Table 2”.
Page 10, line 277 – Page 12, line 314 = This part should be moved at the beginning of the results and discussion to comprehend the work better.
Page 11, Scheme 1: The authors should move the scheme to page 7 before discussing hydrolysed oligosaccharides to comprehend NMR spectra better.
Materials and methods
The authors should state the purchase of all chemicals used in the manuscript, uniform both % concentration as w/v or v/v as done on page 14, line 379, and temperature as K.
Page 13, lines 363= The authors should please indicate which acid has been used and its concentration to adjust the pH to 2.8.
Page 14, line 414= The authors should please indicate which base has been used and its concentration to adjust the pH to 7.0.
Page 15, NMR analysis = The author should state the experimental conditions of NMR acquisitions.
Author Response
Dear reviewer,
We would like to show our great appreciation for your comments and suggestions on our manuscript. The manuscript has been modified accordingly, and detailed corrections are listed below point by point:
With a relative anticoagulant activity evaluation, the present study exploits the structural characterisation of polysaccharide repeating units of sulphated fucan extracted from Holothuria floridana.
The authors should revise the English and measurement units in the main paper for the presence of many typos, repetitions, and redundancy. The manuscript is acceptable for publishing after major revisions, as it lacks unity, and the results are confusedly described. It seems to be written by many hands.
Thanks very much for the reviewer’s comments. We apologize for the grammatical problems and have revised the WHOLE manuscript according to your suggestions. In addition, we will apply English language editing in the next step.
Moreover, several issues should be considered in graphs and modified carefully by the authors.
There are specific problems as follows:
Introduction:
Page 2, lines 45-49 = The authors should type the name of the invertebrates with their full names since they are introduced for the first time. Moreover, they should describe the chemical structure of the repeating unit, as done for H. Floridana.
Thanks for your suggestion. The full names of the sea cucumbers have been provided in our revised manuscript. The chemical structure of the repeating unit in the FS isolated from these sea cucumbers have been showed in Scheme 1.
Page 2, lines 57-64 = The authors should report state of the art for the several methods used for the depolymerisation of fucans and their effect on the sulfation pattern of polysaccharides.
Generally, enzymatic hydrolysis, peroxidative depolymerization, and acid hydrolysis have been described for the depolymerization of FS. Peroxidative depolymerization is a common method to obtain low-molecular weight product from native polysaccharides. The chemical compositions of the product could be consistent with that of native one during the depolymerization. Recently, the structures of a fucan sulfate from sea cucumber Stichopus herrmanni and its depolymerized products prepared by peroxidative degradation were studied extensively. The results indicated that peroxidative degradation involved in cleavage of glycosidic bonds and oxidative modification of reducing end of sugar residue, while showed no destruction on the basic structure, and no sulfate loss was observed.
Mild acid hydrolysis is considered as a non-specific method for polysaccharide de-polymerization, which has been used in FS from sea urchins. The results indicated that the glycosidic linkage between the non-sulfated fucose unit and the adjacent unit was preferentially cleaved by acid, and the selective 2-desulfation was observed. With regard to the effect of mild acid hydrolysis on the sulfate content of fucan sulfate, the sulfate content of dHfFS-2 was determined by ion chromatography to be 31.8%. The result was similar to that of native FS (30.3%), suggesting that the mild acid hydrolysis conditions used in our study did not cause obvious desulfation.
Page 2, line 75 = Please use round brackets for glycosidic bonds in the structural formula.
Thanks for your suggestion. We have added the round brackets for glycosidic bonds in the structural formula in our revised manuscript.
Page 2, line 81 = The authors report the use of free radical depolymerisation, which should also be stated in the abstract. Please uniform.
Thanks for your suggestion. We have stated the free radical depolymerization in the abstract section (lines 16-17).
Results and discussion:
Page 2, lines 94-95 = The authors should move “Shodex OH-pak SB-804 HQ column (8 mm x 300 mm)” in the “Materials and methods” section.
“Shodex OH-pak SB-804 HQ column (8 mm × 300 mm)” has been deleted in lines 94-95 and showed in the “Materials and methods” section (line 458).
Page 3, Figure 1 = The authors, after, HPGPC chromatograms, describe in the main text HPLC ones; therefore, they should give consistency and uniformity, naming HPLC chromatograms as B, then IR as C and so on.
We are very sorry for our negligence of the order of the Figure 1 cited in the main text, and we have corrected the Figure 1.
“Figure 1. HPGPC of crude polysaccharide, native HfFS and dHfFS-1 (A), HPLC profiles of PMP derivatives of mixed monosaccharide standards and HfFS (B), sulfate elution profiles of HfFS obtained using ion chromatography (C), and IR spectrum of HfFS (D).”
Page 3, Figure 1 A = The authors should use different colors for different curves.
Figure 1A has been revised and different curves have been showed in different colors.
Page 3, Figure 1 D = The authors should zoom the calibration curve and preferably move to the "Supporting Info".
Thanks for your suggestion, and the calibration curve has been showed in Supporting information (Figure S1).
Page 3, lines 119-126 = The authors should reorder the discussion about bands assignment, starting from 4000 to 400 cm-1. Moreover, please assign all the bands detected.
Thanks for your suggestion. The observed bands have been assigned starting from 4000 to 400 cm-1 in our revised manuscript.
Page 4, line 131 = Please state “red curve” in Figure 2 A.
The “red curve” in Figure 2A was the 1H NMR spectrum of the peroxidative depolymerized HfFS (dHfFS-1).
Page 5, Figure 2 B = The authors display this figure without describing it in the main text. They only report the C-5 chemical shift of dHfFS-1. Please remove the figure or discuss it in the text.
Thanks for the reviewer’s suggestion. We changed the order of graphs in Figure 2 and added the description of 13C NMR spectrum of dHfFS-1.
Page 5, Figure 2 C = Please change the figure with the un-zoomed one for more evident evidence of correlations.
Thanks for your suggestion. The un-zoomed overlapped 1H-1H COSY (gray), TOCSY (red) and ROESY (green) spectra was provided in Supporting information (Figure S2).
Page 5, line 170 = The authors should move “Shodex OH-pak SB-804 HQ column (8 mm x 300 mm)” in the “Materials and methods” section.
“Shodex OH-pak SB-804 HQ column (8 mm × 300 mm)” has been deleted in line 170.
Page 5, Paragraph 2.4 = The authors state the mild acid hydrolysis at 50°C, but the obtained result is not discussed, differently from the one conducted at 80°C.
Thanks for the reviewer’s suggestion. We have rewritten this paragraph in the revised manuscript.
Page 6, line 194 = The authors report the F1 fraction, which is a mixture of oligosaccharides; therefore, they should state and discuss F1-a and F1-b, as written in Scheme 1.
Thanks for your suggestion. The two components in F1 fraction have been specified as F1-a and F1-b the main text.
Page 6, lines 196-197 = The authors should discuss NMR spectra reported in Figure S1, as done for other fractions.
Thanks for your professional suggestion. We have added a paragraph to discuss the NMR spectra in Figure S3 in our revised manuscript.
Page 7, line 208 = Please change “Table 1” with “Table 2”.
“Table 1” in line 208 has been changed with “Table 2”.
Page 10, line 277 – Page 12, line 314 = This part should be moved at the beginning of the results and discussion to comprehend the work better.
Thanks for your suggestion, and the mentioned main text has been showed at the beginning of the results and discussion.
Page 11, Scheme 1: The authors should move the scheme to page 7 before discussing hydrolysed oligosaccharides to comprehend NMR spectra better.
Thanks for your suggestion, and Scheme 1 has been moved to page 7 before discussing the structure of oligosaccharides.
Materials and methods
The authors should state the purchase of all chemicals used in the manuscript, uniform both % concentration as w/v or v/v as done on page 14, line 379, and temperature as K.
Thanks for your suggestion. All chemicals used in the manuscript were obtained commercially. The % concentration has been labeled as w/v or v/v in our revised manuscript.
Page 13, lines 363= The authors should please indicate which acid has been used and its concentration to adjust the pH to 2.8.
6 M HCl was used to adjust the pH to 2.8.
Page 14, line 414= The authors should please indicate which base has been used and its concentration to adjust the pH to 7.0.
6 M NaOH was used to adjust the pH to 7.0.
Page 15, NMR analysis = The author should state the experimental conditions of NMR acquisitions.
Thanks for your suggestion. The detailed experimental conditions of NMR acquisitions have been shown in Supporting information.
Reviewer 3 Report
In the manuscript of Z. Ning, et al. “A fucan sulfate with pentasaccharide repeating units from the sea cucumber Holothuria floridana and its anticoagulant activity", a fucan sulfate (abbreviated as HfFS) was isolated from the sea cucumber Holothuria floridana after proteolysis-alkaline treatment and purified by anion-exchange chromatography. The molecular weight of HfFS was determined as 443.4 kDa and the molar ratio of sulfate ester to fucose was found as 1.11:1. In HfFS, the authors revealed pentasaccharide repeating unit –[L-Fuc2S4S-α-(1-3)-L-Fuc-α-(1-3)-Fuc-α-(1-3)-L-Fuc2S-α-(1-3)-L-Fuc2S-α-(1-3)-]n-. The “bottom-up” strategy was used to confirm the structure of HfFS: a series of fucooligosaccharides (disaccharides, trisaccharides, and tetrasaccharides) were obtained by the mild acid hydrolysis of HfFS. Their structures were evaluated from 1D and 2D NMR spectra and ESI HRMS. Anticoagulant activity assays of native HfFS and its depolymerized products (dHf-1~dHf-6) in vitro suggested that HfFS and related oligosaccharides exhibit potent APTT activity decreased with the reduction of molecular weight: HfFS fragments (dHf-4-6) with Mw less than 11.5 kDa showed no significant anticoagulant effect. The structure of HfFS was compared with other FSs isolated from different sea cucumbers.
Reviewer's notes. 1. Lines 124—125, 2.2 Infrared (IR) spectroscopy analysis. Peaks at 1458, 1390, and 1355 cm–1 are not matched in Figure 1B.
- Supplementary Materials, Table S2. The column "Predicted Peak": for all m/z values, mass of an electron, 5.5 mDa is missed. Please, correct these calculations. ESI HRMS conditions are not described in Materials and Methods, lines 347 and below.
- Lines 278—290. The paper N.E, Ustyuzhanina, M.I. Bilan, A.S. Dmitrenok, E.Y. Borodina, N.E. Nifantiev, A.I. Usov, A highly regular fucan sulfate from sea cucumber Stichopus horrens, Carbohydr. Res., 2018, 456, 5—9 is not discussed. Please, add this reference.
Author Response
Dear reviewer,
We would like to show our great appreciation for your comments and suggestions on our manuscript. The manuscript has been modified accordingly, and detailed corrections are listed below point by point:
Comments and Suggestions for Authors
In the manuscript of Z. Ning, et al. “A fucan sulfate with pentasaccharide repeating units from the sea cucumber Holothuria floridana and its anticoagulant activity", a fucan sulfate (abbreviated as HfFS) was isolated from the sea cucumber Holothuria floridana after proteolysis-alkaline treatment and purified by anion-exchange chromatography. The molecular weight of HfFS was determined as 443.4 kDa and the molar ratio of sulfate ester to fucose was found as 1.11:1. In HfFS, the authors revealed pentasaccharide repeating unit –[L-Fuc2S4S-α-(1-3)-L-Fuc-α-(1-3)-Fuc-α-(1-3)-L-Fuc2S-α-(1-3)-L-Fuc2S-α-(1-3)-]n-. The “bottom-up” strategy was used to confirm the structure of HfFS: a series of fucooligosaccharides (disaccharides, trisaccharides, and tetrasaccharides) were obtained by the mild acid hydrolysis of HfFS. Their structures were evaluated from 1D and 2D NMR spectra and ESI HRMS. Anticoagulant activity assays of native HfFS and its depolymerized products (dHf-1~dHf-6) in vitro suggested that HfFS and related oligosaccharides exhibit potent APTT activity decreased with the reduction of molecular weight: HfFS fragments (dHf-4-6) with Mw less than 11.5 kDa showed no significant anticoagulant effect. The structure of HfFS was compared with other FSs isolated from different sea cucumbers.
Reviewer's notes. 1. Lines 124—125, 2.2 Infrared (IR) spectroscopy analysis. Peaks at 1458, 1390, and 1355 cm–1 are not matched in Figure 1B.
Thanks for your comments on our manuscript. We have marked these peaks in the IR spectrum Figure 1.
Supplementary Materials, Table S2. The column "Predicted Peak": for all m/z values, mass of an electron, 5.5 mDa is missed. Please, correct these calculations. ESI HRMS conditions are not described in Materials and Methods, lines 347 and below.
Thanks for your careful review. We have revised the “Predicted Peak” in Table S2 and the conditions for acquiring the MS spectra of oligosaccharides in Section 3.9 in Materials and Methods.
Lines 278—290. The paper N.E, Ustyuzhanina, M.I. Bilan, A.S. Dmitrenok, E.Y. Borodina, N.E. Nifantiev, A.I. Usov, A highly regular fucan sulfate from sea cucumber Stichopus horrens, Carbohydr. Res., 2018, 456, 5—9 is not discussed. Please, add this reference.
Thanks for your suggestion, and the reference has been added and discussed in our revised manuscript.
Reviewer 4 Report
This manuscript reports a fucan sulfate with pentasaccharide repeating units from the sea cucumber and its anticoagulant activity. This is an interesting study and the information could be useful in the research of glycan based biomaterial.
Comments:
- What is the mechanism of the anticoagulant activity from the fucan sulfate with pentasaccharide repeating units?
- It is necessary to test the molecular interactions between fucan sulfate with some key proteins (such as ATIII) related to anticoagulant activity?
- If fucan sulfate is used as an anticoagulant in human, how is the toxicity and biodegradability in vivo?
Author Response
Dear reviewer,
We would like to show our great appreciation for your comments and questions on our manuscript. We discussed the questions point by point:
This manuscript reports a fucan sulfate with pentasaccharide repeating units from the sea cucumber and its anticoagulant activity. This is an interesting study and the information could be useful in the research of glycan based biomaterial.
Comments:
What is the mechanism of the anticoagulant activity from the fucan sulfate with pentasaccharide repeating units?
We thank the reviewer’s good question. In fact, we investigated the anticoagulant mechanism of fucan sulfate (FS) from several sea cucumber species. For example, we investigated the anticoagulant mechanism of FS from the sea cucumbers Holothuria edulis and Ludwigothurea grisea by chromogenic substrate method [1]. The anti-thrombin activity in the presence of heparin cofactor II, anti-factor Xa and anti-thrombin activities mediated by AT were determined. The results indicated that the FS could selectively inhibit thrombin activity in the presence of heparin cofactor II, while displayed weak anti-thrombin activities mediated by AT and negligible anti-factor Xa activities. The previous results could provide meaningful reference for the possible anticoagulant mechanism of HfFS. However, the structures of fucans may vary from species to species, so this may give rise to variation in the detailed mechanisms of anticoagulant action. For example, a study of the FS from sea urchin suggested that[2], the occurrence of 2,4-di-O-sulfated units is an amplifying block for 3-linked L-fucan enhanced thrombin inhibition by AT, and the major structural requirement for anti-thrombin activity by heparin cofactor II should be related to the 4-O-sulfated L-fucose units. Overall, the anticoagulant activity and mechanism should be related to the structure characteristics of FS. In this study, our results indicated that the anticoagulant activity of HfFS with pentasaccharide repeating unit was dependent to its Mw, and the detailed mechanism will be discussed in our future work.
It is necessary to test the molecular interactions between fucan sulfate with some key proteins (such as ATIII) related to anticoagulant activity?
Thanks for your insightful advice. As our response to the reviewer’s last question, refer to our previous work [1], the FS from the sea cucumbers Holothuria edulis and Ludwigothurea grisea could selectively inhibit thrombin activity in the presence of heparin cofactor II, while displayed weak anti-thrombin activities mediated by AT and negligible anti-factor Xa activities. The molecular interactions between FS and some key proteins (such as ATIII, HC-II) are important data to indicate the anticoagulant mechanism. We have carefully evaluated the experimental conditions required to complete the work, and feel that we cannot afford this expanded scope of supplementary research at this time. At the same time, the current manuscript mainly focuses on the structure elucidation of FS from the sea cucumber Holothuria floridana, and the anticoagulant activity assays suggest that the Mw of HfFS plays an important role in the anticoagulant activity. Based on the results in the present manuscript, we will prepare the low-molecular-weight HfFS product with Mw of ~20 kDa, which will be used for further anticoagulant mechanism studies and the results will be included in another follow-up paper.
If fucan sulfate is used as an anticoagulant in human, how is the toxicity and biodegradability in vivo?
This is a thoughtful question. We provide some discussion about the above concern based on references. Up to date, fucan sulfate has not yet been developed as an anticoagulant in human, but scientists believe that it has good prospects of being developed as an anticoagulant. For example, the effects of FS from the sea cucumber Ludwigothurea grisea on coagulation, thrombosis and bleeding were studied both in vitro and in vivo [3]. This FS could inhibit venous thrombosis in rats at low doses (up to 1.0 mg/kg body weight). Several studies identified FS from marine invertebrates or algae are non-toxic for the tested cells in vitro [4]. However, the toxicity of FS in vivo needs further investigation.
The antithrombosis activity of FS was usually evaluated after administering intravenously or subcutaneously. Although few studies investigated the biodegradability of FS according to our knowledge, our group reported the excretion of dHG-5 in rats, a low-molecular-weight product of the fucosylated glycosaminoglycan from sea cucumber, and the results indicated that dHG-5 was mainly excreted by urine as the unchanged parent drug and about 60% was excreted within 48 h for s.c. and i.v. administration [5]. No obvious biodegradability of dHG-5 was observed. Additionally, the bioavailability and biodegradability of FS by oral administration may be largely unknown. According to the references [6], FS may be degraded by gut microbiota, and the degradation ability is related to the chemical structure, which needs further studies. However, your comments provide us with novel research interests.
- Wu, M.; Xu, L.; Zhao, L.; Xiao, C.; Gao, N.; Luo, L.; Yang, L.; Li, Z.; Chen, L.; Zhao, J., Structural analysis and anticoagulant activities of the novel sulfated fucan possessing a regular well-defined repeating unit from sea cucumber. Marine Drugs 2015, 13, (4), 2063-84.
- Pereira, M.S.; Melo, F.R.; Mourão, P.A.S. Is there a correlation between structure and anticoagulant action of sulfated galactans and sulfated fucans. Glycobiology 2002, 12, 573-580.
- Fonseca, R. J. C.; Santos, G. R. C.; Mourão, P. A. S., Effects of polysaccharides enriched in 2,4-disulfated fucose units on coagulation, thrombosis and bleeding. Thromb Haemost 2009, 102, (11), 829-836.
- Usoltseva, R. V.; Shevchenko, N. M.; Silchenko, A. S.; Anastyuk, S. D.; Zvyagintsev, N. V.; Ermakova, S. P., Determination of the structure and in vitro anticancer activity of fucan from Saccharina dentigera and its derivatives. International Journal of Biological Macromolecules 2022, 206, 614-620.
- Liu, S.; Zhang, T.; Sun, H.; Lin, L.; Gao, N.; Wang, W.; Li, S.; Zhao, J., Pharmacokinetics and pharmacodynamics of a depolymerized glycosaminoglycan from Holothuria fuscopunctata, a novel anticoagulant candidate, in rats by bioanalytical methods. Marine Drugs 2021, 19, (4), 212.
- Wei, B.; Zhang, B.; Du, A.; Zhou, Z.; Lu, D.; Zhu, Z.; Ke, S.; Wang, S.; Yu, Y.; Chen, J.; Zhang, H.; Jin, W.; Wang, H., Saccharina japonica fucan suppresses high fat diet-induced obesity and enriches fucoidan-degrading gut bacteria. Carbohydrate Polymers 2022, 290, 119411.
Round 2
Reviewer 1 Report
This paper has been improved greatly. You have addressed the issue of the sulfate content of depolymerised polysaccharide in your response to my comments, but I can see no mention of this in the revised manuscript. I think it is important to state that depolymerisation did not alter the sulfate ratio and show the data.
Other comments:
Line 139: resonances at 1.2-1.4 ppm are H6 protons not H5.
Author Response
Dear reviewer,
We would like to show our great appreciation for your comments and suggestions on our manuscript. The manuscript has been modified accordingly, and detailed corrections are listed below point by point:
This paper has been improved greatly. You have addressed the issue of the sulfate content of depolymerised polysaccharide in your response to my comments, but I can see no mention of this in the revised manuscript. I think it is important to state that depolymerisation did not alter the sulfate ratio and show the data.
We sincerely appreciate the valuable comments on our manuscript. The result of the sulfate content of dHfFS-2 has been added in Supporting information (Figure S1) and described in the main text in our revised manuscript (line 181-184).
Other comments:
Line 139: resonances at 1.2-1.4 ppm are H6 protons not H5.
Thanks for your suggestion, and we have revised this sentence.
Reviewer 2 Report
The authors have modified the manuscript accordingly to the reviewer's suggestion. Therefore, it is could be accepted in the present form.
Author Response
We would like to show our great appreciation for your positive comments on our paper.